# Advances in the Preparation of Nanofiber Dressings by Electrospinning for Promoting Diabetic Wound Healing

**DOI:** 10.3390/biom12121727

**Published:** 2022-11-22

**Authors:** Yukang Liu, Chaofei Li, Zhangbin Feng, Biao Han, Deng-Guang Yu, Ke Wang

**Affiliations:** 1School of Materials and Chemistry, University of Shanghai for Science and Technology, Shanghai 200093, China; 2Department of General Surgery, Ruijin Hospital Affiliated to Shanghai Jiao Tong University School of Medicine, Shanghai 200025, China

**Keywords:** electrospinning, nanofiber, diabetic wound, wound dressing, nanostructures

## Abstract

Chronic diabetic wounds are one of the main complications of diabetes, manifested by persistent inflammation, decreased epithelialization motility, and impaired wound healing. This will not only lead to the repeated hospitalization of patients, but also bear expensive hospitalization costs. In severe cases, it can lead to amputation, sepsis or death. Electrospun nanofibers membranes have the characteristics of high porosity, high specific surface area, and easy functionalization of structure, so they can be used as a safe and effective platform in the treatment of diabetic wounds and have great application potential. This article briefly reviewed the pathogenesis of chronic diabetic wounds and the types of dressings commonly used, and then reviewed the development of electrospinning technology in recent years and the advantages of electrospun nanofibers in the treatment of diabetic wounds. Finally, the reports of different types of nanofiber dressings on diabetic wounds are summarized, and the method of using multi-drug combination therapy in diabetic wounds is emphasized, which provides new ideas for the effective treatment of diabetic wounds.

## 1. Introduction

The skin is the largest organ of the human body, accounting for about 15% of the body mass. In human life, the skin has always played an important role in protecting the body and maintaining normal physiological functions [1,2]. However, various internal and external factors, such as external physical forces, surgery, burns, and chemical damage, as well as chronic trauma caused by diabetes, can cause varying degrees of damage to the skin [3,4]. A wound formation is recognized when the integrity of the skin is compromised or disrupted, including abrasions, surface burns, diabetic wounds, and lower-extremity venous ulcers [5]. Wound healing is the process of repairing and remodeling tissue structural damage. Tissue repair is the process of replacing injured tissue with other tissues for healing, such as fibrosis or scarring, which generally makes the damaged tissue lose its biological function. In contrast, regeneration is to replace the damaged tissue with the same cells, so that the biological function of the injured tissue can be restored [6]. Wound healing consists of four main phases: hemostasis, inflammation, proliferation, and remodeling [7,8]. These stages involve a wide range of cell types, extracellular compositions, growth factors, and cytokines, and diabetes causes impaired wound healing by affecting one or more of these biological mechanisms. Chronic, non-healing diabetic wounds may lead to repeated hospitalizations, often worsening the condition and leading to amputation of the affected limb [9,10,11].

To date, the gold-standard treatment for diabetic foot ulcers is debridement, glycemic control, infection control, decompression, moist dressings, and orthotic use. Debridement is a process for removing non-viable and necrotic tissue. That is, the removal of callus, necrotic tissue and bacterial components at the wound by surgery, hydrogel, biosurgery (i.e., maggots), to promote the release of growth factors and accelerate the transformation of chronic wounds into acute wounds [12,13]. In addition, decompression can be achieved by debridement of the healing tissue. Then, a dressing with wet and dry properties is used to maintain a moist wound environment and absorb more than the exudate. Finally, a combination of decompression interventions can minimize the biological stress of the diabetic foot [14].

However, there are significant challenges in developing effective dressings for diabetic wounds. This is because the ideal wound dressing should not only have sustained resistance to infection and the ability to absorb large amounts of wound tissue exudate, but it should also have good gas exchange and provide a moist healing environment for the wound [15]. Diabetic chronic wound treatment is a common medical concern in clinical care, and despite great efforts to manage it, the desired therapeutic effect has still not been achieved. As a result, there is an urgent need for an efficient medication delivery method for topical administration and improved diabetic chronic wound care. Although drug-delivery systems such as films, hydrogels, and ointments have been used in the management of diabetic chronic wounds, nanofibers offer better results in promoting tissue formation, vascular regeneration, and faster wound healing. The porous structure of nanofibers can enhance drug loading and provide continuous drug delivery, making nanofibers widely used in biomedical field [16,17]. To date, nanofibers have been prepared by electrospinning, blow spinning, centrifugal spinning, and microfluidic spinning techniques. Of these, the simple and practical electrostatic spinning technique is widely used to produce fibers in the nano- or micron-diameter range [16,18].

Electrospun nanofiber is a novel material that may be employed for specific purposes in biomedical applications by using electrospinning technology to precisely change the interior structure of the fiber. Due to its good biocompatibility and versatility, it is used in a variety of biomedical applications, such as wound dressings [19,20], tissue engineering [21,22], and drug-delivery systems [23,24], as shown in Figure 1. (1) First, its structural and biological functions are analogous to the natural extracellular matrix, which can act as a cellular scaffold during wound repair, producing physical support for cells and providing the optimal microenvironment for adherence, multiplication, immigration and differentiation [25]. (2) Next, electrospinning fibers have smaller gaps and higher specific surface area, which is beneficial for hemostasis. Furthermore, the higher specific surface area is favorable for liquid adsorption and active ingredient loadings, such as antimicrobial drugs, inorganic nanoparticles, growth factors and extracts of Chinese herbs [26,27]. (3) In addition, nanofibers have high porosity, which facilitates cellular respiration and gas permeation, and prevents wound drying and dehydration. The ideal electrospinning fiber also has an interpenetrating nano-scale pore size, which can block the invasion of external microorganisms [28]. (4) The rates and time for drug releases can be controlled through the adjustment of fiber structure and morphological sizes to promote efficient wound-site healing. Thus, electrospinning fibers exhibit significant potential in the production of diabetic wound dressings.

In order to understand the current research status of electrospinning nanofiber membranes in diabetic wound healing, we searched the Web of Science on the topics of “Electrospinning diabetic wound healing” and “Diabetic wound healing”. The search result is shown in Figure 2. The number of articles published on both topics showed an increasing trend, with the number of articles published on the topic of “Diabetic wound healing” remaining above 1000 for 4 consecutive years from 2017 to 2021, indicating that diabetic wound healing has attracted widespread attention. The number of literature searches on the topic of “Electrospinning diabetic wound healing” has shown a rapid increase in the past 4 years, indicating that electrospinning nanofiber film, as a new type of diabetic wound dressing, has a wide range of promising applications in the healing of diabetic wounds. This article mainly summarizes the advantages and research progress of electrospun nanofibrous membranes in the treatment of diabetic wounds.

## 2. Diabetic Wounds and Diabetic Dressings

### 2.1. Diabetic Wounds

Diabetes mellitus is a medical condition that results from the inability of the pancreas to produce enough insulin or the body’s inability to effectively use the insulin produced, and can be classified as type 1, type 2, and gestational diabetes [29]. Diabetic wounds or foot ulcers (DFU) are a major clinical complication of diabetes [30]. Diabetic wounds are simultaneously chronic, infected and ulcerated wounds characterized by high levels of inflammatory cytokines, impaired angiogenesis, distal arterial arrest and the inability to distribute oxygen, nutrients, and healing cells to the wound site and to perpetuate infection by increasing levels of reactive oxygen species (ROS) [31]. The hemostatic and inflammatory phases of diabetic wounds during healing can proceed normally, but certain internal causes, such as peripheral neuropathy, microcirculatory dysfunction, disruption of growth factor activity and hypoxia [32] make it difficult for diabetic wounds to transition from the inflammatory to the proliferative phase [33].

Chronic diabetic neuropathy, i.e., temporary or permanent nerve tissue damage, is characterized by reduced blood flow and progressive loss of peripheral nerve fibers caused by high blood-glucose levels. Symptoms such as hyperalgesia and autonomic movement disorders may occur during the onset, resulting in abnormal biomechanical changes during the patient’s walking. Autonomic neuropathy can cause a decrease or loss of skin temperature regulation, sweat regulation and blood flow regulation, a decrease in the flexibility of the foot tissue, and the formation of thick calluses, and thick calluses are prone to breakage and cracking, which subsequently increases the risk of infection [34,35].

Most diabetic patients have atherosclerosis, and peripheral vascular abnormalities in the body of diabetic patients are dominated by the arteriopathy of the lower extremities, manifesting soreness and pain in the lower extremities, abnormal sensation, and intermittent claudication, resulting in severe blood supply deficiency, which eventually evolves into diabetic foot [36,37]. Furthermore, low oxygen partial pressure, hyperglycemia, malnutrition, and other factors can cause acid buildup in the foot tissue, hyperosmolarity, and poor anaerobic metabolism, creating an environment that is more favorable to the development and reproduction of pathogenic bacteria [38].

As shown in Figure 3, there is continuous production of large amounts of inflammatory cytokines by neutrophils and macrophages (M1) during the ongoing phase of diabetic inflammation, including tumor necrosis factor-α (TNF-α), Interleukin-1 (IL-1), Interleukin-6 (IL-6) and matrix metalloproteinase (MMP) and ROS. Overexpression of inflammatory factors leads to the rapid degradation of active substances such as cytokines, growth factors, antibiotics, and ECM, which delays stages such as wound healing re-epithelialization, thus interrupting the healing process of diabetic wounds [39,40,41,42]. In addition, pro-inflammatory macrophage “M1” cannot be converted into healing-promoting macrophage “M2”, and the lack of M2 reduces the expression of growth factors and anti-inflammatory cytokines (Interleukin-10 (IL-10)), which also may prolong diabetic wound healing time. Therefore, unregulated and un-intervened diabetic wounds should be of concern in future studies.

### 2.2. Diabetic Dressings

The main objectives of dressing therapy in diabetic wounds are to provide a long-term effective moist and warm environment, as well as to inhibit bacteria, regulate exudation, autolyze debridement, and promote growth and oxygen permeation [43]. As a short-term skin substitute, dressings should be selected based on wound type, sensitivity, amount of exudate and wound depth to aid wound debridement and angiogenesis [44]. The main pathological abnormalities in diabetic wounds are vascular problems and long-term inflammatory response, so single-therapy approaches often fail to achieve the desired results. Studies have shown that wound closure is a closely regulated and highly coordinated process between pro- and anti-inflammation [45]. Therefore, intervention in the healing process to reduce the inflammatory response and stimulate tissue regeneration is currently essential to achieve the faster healing of diabetic wounds [46]. By understanding the characteristics of various dressings, we can easily select wound dressings and simplify the decision-making process. The proper selection of dressings can be of great help in diabetic wound healing. Below is a summary of several commonly used diabetic dressings.

#### 2.2.1. Basic Dressings

Skin trauma is a common clinical problem, and medical dressings are generally used to stop bleeding, reduce inflammation, and promote wound healing. Basic-type dressings have the functions of protective coverage, exudate management, pain relief, reduction of water evaporation, and prevention of infection [12]. In recent years, basic dressings have become an important part of wound healing as the process of wound healing has become better understood. However, due to their lack of anti-infective and growth-promoting properties, they are now mainly used in combination with other therapeutic drugs, or as composite dressing materiasl. (1) Autolytic debridement dressing: Autolytic debridement is a painless and highly selective primitive reaction of the body to eliminate necrotic material, often requiring the application of new dressings to create a moist environment to encourage necrotic tissue autolysis. The accompanying hydrogels and hydrocolloid dressings are widely used in clinical practice. Hydrogel is a wet wound dressing with good moisturizing ability, but it easily leaks when using amorphous hydrogel for debridement, which affects the debridement effect [47]. Hydrocolloids are wound dressings consisting of a cross-linked polymer matrix and an adhesive. The ability to provide a moderate cushioning effect makes it suitable for joint wounds, superficial ulcers and shallow leg ulcers. However, they are also prone to symptoms such as allergic skin reactions or adhesions, and the opaque nature of the dressing limits routine wound examination [48]. (2) Enzymatic debridement dressing: Enzymatic debridement is a method that uses chemical reagents to identify and break down inactivated tissue, usually in smaller wounds and wounds with selective microenvironments that do not require intervention. Enzymatic debridement dressing has the advantages of being cost-effective and efficient [49,50]. (3) Antiseptic dressings: Some research suggests that debridement alone is not sufficient to accelerate chronic wound healing and recommends the use of antiseptics to complement the debridement process and control infection. Antiseptics are a class of drugs that can prevent the growth or damage of microorganisms in living tissues or on the surface of living tissues. Halogenated compounds, alcohol-based agents, and biguanides are often used as antiseptics primarily for chronic wound care because they effectively inhibit bacterial reproduction and promote resistance to the external environment [51,52]. (4) Alginate dressing: Alginate dressing consists of alginate salts, which are highly absorbent and hemostatic and suitable for highly exuding wounds. It is clinically indicated for deep pressure ulcers of the lower extremities, Noma, and exudative ulcers of the lower extremities. However, on wounds with little to no exudate, they adhere to the surface of the healing wound, causing pain and damaging healthy tissue when removed [53]. (5) Polyurethane foams dressings are hydrocolloid dressings with moisturizing, soft, non-toxic and good mechanical properties, and can be used as wound dressings even after complete immersion [54]. In addition, the porous structure inside the polyurethane foam dressing makes the interconnected pores the advantages of good intracellular growth, strong water absorption capacity, and high water-vapor transmission rate. This allows it to effectively absorb exudate and maintain a moist microenvironment on the wound, accelerating wound healing and preventing wound infestation [55].

#### 2.2.2. Antimicrobial Active Dressing

As it may be difficult to achieve the desired effect with oral medication antimicrobial therapy alone in diabetes patients due to poor microcirculation, the use of anti-infective measures on local wounds can help to slow down the progress of infection and help wound healing [56]. (1) Silver is considered a broad-spectrum antimicrobial agent for superficially infected wounds. The application of this dressing on the skin prevents bacterial invasion of the wound and protects the tissue from the damage to the wound surface. In addition, silver-containing dressings can improve short-term wounds and ulcers [57]. Xu et al. [58], observed the clinical efficacy of silver ion dressing combined with polyhexamethylenebiguanide (PHBM) dressing, and the results showed that the total efficiency of silver ions combined with PHBM for the therapy of DFU infected wounds was significantly higher than that of the PHBM alone control group, which not only had a strong antibacterial effect, but also reduced the inflammatory reaction of the ulcer surface, promoted the growth of granulation tissue, shortened the therapy time, and improved the prognosis. (2) Medical honey dressing: Honey has the ability to inhibit more than 50 types of bacteria, but no microbial resistance has been found [59]. Medical honey also promotes the self-soluble removal of residual substances from wounds. Honey includes nutrients that are structurally comparable to those found in human tissues, which can help wounds heal faster [60]. (3) Iodine-based dressings: Iodine-based dressing is a polymeric material containing iodine, whose main components are iodine and gelatin and other macromolecular substances. Iodine-based dressings can effectively clear wounds and kill bacteria. In addition, there are no reports on the drug resistance of iodine-based dressings compared with antibiotics and antibacterial drugs used in clinical practice [61].

#### 2.2.3. Chinese Medicine Poultice Dressing

Chinese medicine poultice is made into various plasters composed of various prescriptions and in different dosage forms according to the location of the disease and the different types of identification. It is a characteristic application of traditional Chinese medical surgical therapy. Chinese medicine plaster dressing is suitable for various stages of surgical sores, pus formation, and ulcers. It is soft in texture, without the discomfort of hard adhesion, and can be used for tears and creases of wounds, or large-scale ulcers.

Multiple studies have reported the use of traditional Chinese medicine plaster for DFU wound healing. Li et al. [62], applied “SheXiangYuHong” blood healing ointment to wounds of diabetic mice to observe the effect of the ointment in the healing stage of diabetic mice. The analysis revealed a considerable rise in the pace of granulation of the wounds to which the ointment was administered, as well as a significant improvement in the deposition of collagen fibers and the formation of tissue on the wounds. On the other hand, “SheXiangYuHong” cream can also regulate the expression of Interleukin-6 (IL-6) [63] and tumor necrosis factor-α (TNF-α) [64], which are pro-inflammatory factors involved in the inflammatory process, to slow down the inflammatory response. Fei et al. [65], discussed the effects of Shixiang plaster and aminoguanidine on wound healing in diabetic rats. Shixiang plaster is a traditional Chinese medicine commonly used to treat chronic wounds of diabetic ulcers. Aminoguanidine is a hydrazine compound that inhibits the expression of advanced glycosylation end products (AGEs) [66]. Shixiang plaster showed a significant increase in granulation tissue and angiogenesis rates and a significant decrease in AGEs in the treated group compared to the untreated group with chronic ulcers. Therefore, Chinese medicine plaster has a promising application in diabetic wound therapy.

#### 2.2.4. Bioactive Dressings

The bioactive dressing is a novel wound repair and protection dressing developed on the basis of the “moist healing environment theory”. Most of these dressings are composites based on natural and synthetic polymers, with the addition of antimicrobial drugs, growth factors, and inorganic nanoparticles to access antimicrobial activity and to facilitate fibroblast activation and migration of vascular endothelial cells [67]. These dressings are constructed of a variety of polymers, including gelatin, silk, chitosan, sodium alginate, and employed in forms of nanofibers, films, foams, sponges and hydrogels [68,69].

Nanofiber wound dressings with an average diameter of less than 1 micron are applied to the injured area to preserve the shape and function of the wound. It is also easily removed when dressings are replaced. Nanofiber dressings have been widely investigated because of their low surface tension, flexibility and biocompatibility compared with traditional materials [70,71]. Electrospinning is one of the most commonly used application techniques for preparing nanofibers, and the fiber membranes prepared by this method have the advantages of large specific surface area, high porosity, tunable mechanical properties and high cost-effectiveness [72]. Compared to other wound dressings, electrospun dressings allow for the faster delivery of drugs or active substances to the wound, do not require frequent dressing changes, and have an inherent potential to heal wounds [73,74]. To date, the electrospinning of nanofibrous membranes is the most practical technique in the field of advanced dressings for tissue engineering and the delivery of bioactive agents to damaged tissues and has attracted a lot of attention from researchers [75].

## 3. Electrospinning

### 3.1. Electrospinning Technology

As a “top-down” fiber-production technique, electrospinning has the advantages of simple operation, low production cost and low energy consumption. This method can effectively combine polymer materials with inorganic nanoparticles to produce nano- or micron-diameter fibers in a relatively short time [76,77]. The earliest and most commonly used system in electrospinning is a single spinneret system. Later, with the continuous progress of electrospinning technology and different needs, electrospinning systems with double needles and multi-needles have been gradually developed. As research on electrospinning continues to deepen, nanofibers with specific shapes and structures are tailored by varying parameters such as polymer type, molecular weight, solution conductivity, voltage, flow rate, and temperature and humidity [78,79]. For instance, porous [80], pleated [81], hollow [82], beaded [83,84], and reticulated [85] structures. Therefore, electrospinning has become a hot research direction in major fields.

### 3.2. Classification of Electrospinning Process

#### 3.2.1. Single-Fluid Electrospinning

Single-fluid electrospinning, also known as hybrid electrospinning, prepares nanofibers by mixing drugs and polymers directly from a single spinneret. The traditional single-fluid electrospinning technique has attracted tremendous interest from researchers over the years owing to its simple experimental device, lower cost, higher throughput, and ease of manipulation [20,86]. However, hybrid electrospinning has major limitations on the materials prepared—the polymers must be spinnable, which largely limits the development of the process. In addition, if the drug mixture has low solubility, the drug would be loaded on the fiber face, resulting in a drug burst release situation [87,88].

#### 3.2.2. Multifluid Electrospinning

Due to the numerous limitations of single-fluid electrospinning, it also lacks the ability to fabricate complex hierarchical nanotubes or side-by-side structures. Therefore, multifluid electrospinning is derived, that is, electrospinning has two or more working fluids [89,90]. At present, the common multifluid electrospinning process mainly has two-fluid (coaxial electrospinning, side-by-side electrospinning), three-fluid (tri-layer coaxial, tri-layer side-by-side), and other special structures, etc. [23,91,92,93], as shown in Figure 4C,D.

##### Coaxial Electrospinning

Coaxial electrospinning is an electrospinning process that uses two spinnerets to prepare fibers with a core-sheath structure, which is composed of two nested needles of different calibers in a concentric circle structure, as shown in Figure 4C. This process enables the preparation of one-dimensional nanomaterials with core-sheath structures by the direct electrospinning of two polymer solutions without direct mixing. Core-sheath-structured fibers have numerous advantages, especially in the construction of long-lasting controlled-release drug-delivery applications. (1) Load the drug into the core layer, and effectively protect the drug from the influence of the external environment through the shell layer [94]. (2) The composition and thickness of the core-sheath layer fibers are adjusted to achieve the function of controlling the slow release of the drug [95]. (3) To prepare a multi-stage multi-effect drug-release system by encapsulating different drug molecules in the core-sheath layer separately [96]. Liu et al. [97], prepared a core-sheath nanofiber by coaxial electrospinning that was able to control both the site of drug release and the retarded release of the drug. In vitro dissolution showed that aspirin was released only 8.17% ± 3.4% in the environment of artificial gastric juice for 2 h, showing a sustained release in the subsequent time. Liu et al. [98], prepared core-sheath nanofibers using hydrophilic polyvinylpyrrolidone (PVP) as a drug carrier and hydrophobic poly (3-hydroxybutyric acid-co-3-hydroxyvaleric acid) (PHBV) as a protective layer. The core-sheath fibers not only increased the solubility of curcumin but also enabled the slow-release distribution of its drug over 24 h.

##### Side-by-Side Electrospinning

Side-by-side electrospinning is the combination of two spinnerets side-by-side to prepare composite fibers consisting of two materials. In contrast to core-sheath nanofibers, the chambers on both sides of the side-by-side structure are separated from each other and both can be in direct contact with the environment. This trait makes it more popular in the use of functional nanomaterials and offers a solid foundation for the development of complicated structure–property–utility interactions for innovative nanomaterials [83]. However, due to the difficulty of dealing with multiple working fluids at the same time and the reasonable design of the spinning head, there are few studies on this issue.

The conventional side-by-side structure and the modified side-by-side spinneret structure are represented in the first two parts of Figure 4C. Wang et al. [99], fabricated Janus fibers loaded with LO and Ag nanoparticles using a conventional side-by-side electrospinning process, as shown in Figure 5. The prepared fiber membranes have excellent hydrophobic properties and synergistic antibacterial properties. The modified side-by-side technology has a round side outlet that is somewhat higher than the crescent side outlet, allowing the charge to be dispersed equally across the round line on the exterior of the metal capillary. Furthermore, sufficient contact area may overcome the disadvantages of phase separation owing to material rejection in conventional side-by-side electrospinning, ensuring that side-by-side nanofibers are prepared effectively [100]. Yang et al. [101], first used this modified eccentric spinneret electrospinning to successfully prepare a trauma dressing of polyvinylpyrrolidone and ethylcellulose loaded with ciprofloxacin (CIP) and AgNPs heterogeneous Janus nanofibers, respectively, which showed good performance in both controlled drug-release and antibacterial tests, opening a new way to create advanced functional nanomaterials based on Janus structures. Xu et al. [100], used electrospinning to prepare hierarchically structured trauma dressings with drug-loaded monolithic fibers and side-by-side nanofibers, as shown in Figure 5. The composite intermediate layer was Janus nanofibers prepared from PCL//Gel as a matrix loaded with CIP and ZnONPs to ensure initial drug release and sustained synergistic antimicrobial effect in trauma therapy.

##### Multifluid Electrospinning

Triaxial electrospinning is an electrospinning technique for making tri-layer nanofibers using three distinct working fluids. Triaxial electrospinning technology has advanced rapidly during the previous two decades. Triaxial electrospinning spinnerets come in a variety of structural designs, as shown in Figure 4D [90,92,102]. Researchers have created a range of composite nanofibers based on polymers doped with different functional particles or nanoparticles using this approach.

Yu et al. [103], successfully established a standard triaxial coaxial electrospinning technique in 2015 utilizing a homemade the tri-layer concentric spinneret, which can regulate the electrospinning of three fluids at the same time to manufacture nanofibers with a tri-layer gradient structure and provide graded uniform drug-release control. Zhao et al. [104], produced core-sheath fiber membranes for antibiotic breakdown using modified triaxial electrospinning loaded with functional particles. The results demonstrated that under the influence of natural light, the functional nanofiber membranes could break down doxycycline effectively. Therefore, the production of core-sheath nanofibers by three-stage coaxial electrospinning technology has great application prospects in the preparation of novel functional nanomaterials.

## 4. Electrospinning Nanofiber in Diabetic Wound Therapy

Nanofibers created utilizing electrospinning technology have shown outstanding wound healing properties. Their architecture closely resembles the human ECM structure and may be used as a dressing to promote cell adhesion and proliferation, eventually prompting new skin regeneration [105,106]. Simultaneously, the high permeability and absorption rate may absorb the exudate created on the trauma site and keep the healing environment moist [107,108]. Consequently, electrospinning nanofibrous material is regarded as the ideal option for advanced wound dressings today [109,110].

Polymer wound dressings exhibit numerous excellent properties that promote chronic wound healing and have attracted great interest in the treatment of diabetic wounds [46,111]. The following mainly summarizes the application of natural polymers, synthetic polymers and their blends to prepare diabetic wound dressings. It also emphasized the advantages of combining multiple drugs in the therapy of diabetic trauma. Table 1 summarizes the studies on the use of nanofiber dressings prepared from different polymers for diabetic wound treatment.

### 4.1. Natural Polymer Electrospinning Fiber for Diabetic Wounds

Natural polymers are available from microorganisms, animals and plants, and often possess protein or polysaccharide properties. These polymers exhibit excellent properties in wound repair; for instance, they can maintain stable tissue structure, provide excellent nutrient supply, and reduce inflammation and infection risk. In addition, the high retention of water maintains a moist wound environment and enhances wound healing [139,140]. The following are a couple of typically natural polymers that have been widely employed in the treatment of diabetic wounds.

Cellulose acetate (CA), the commonest type of cellulose derivative, can absorb large amounts of exudate from diabetic wounds and maintain a moist environment due to its water-absorbing and swelling properties, thereby improving wound healing [141,142,143].

Hyaluronic acid (HA) is an important glycosaminoglycan that acts as a major cytoplasmic constituent and is usually widely distributed as a high-molecular-weight polymer in connective tissue, synovial fluid and epithelial cells. Due to its low immunogenicity, excellent biocompatibility and high water-retention capacity, it is extensively used as a raw material for surgical skincare, such as wound dressings, films and hydrogels [144]. HA can be involved in the three phases of wound healing hemostasis, inflammation, and epithelial regeneration, and high-molecular-weight HA can promote the transition of macrophages from a pro-inflammatory M1 to a reparative M2 phenotype. Macrophage M2 reduces the inflammatory response by releasing anti-inflammatory factors and growth factors, thereby enhancing cell proliferation [145,146,147].

Chitosan (CS), a natural polysaccharide derived from chitin, is considered an ideal biomaterial for wound repair because of its unique cationic antibacterial properties and its naturally occurring bioactivity and excellent biocompatibility. Chitosan can replace glycosaminoglycan in ECM and exert coagulation and hemostasis and promote the infiltration and migration of macrophages during the initial healing phase. In the later healing phase, CS ensures effective wound healing by stimulating epithelial tissue regeneration and reducing scar production [148]. However, CS has rather poor rheological properties and tends to swell in water, which makes it structurally too weak to be used alone in any application [149,150].

Samadian [112] used electrospinning technology to prepare a DFU dressing containing cellulose acetate CA blended with gelatin nanofibers. The prepared fibrous membrane has no adverse effect on the cultured cells. Collagen production density was found to be 88.8% and angiogenesis was found to be 19.8% in animal studies. The above results indicated that loading berberine into CA/gelatin nanofiber dressings not only did not affect the physicochemical properties of the dressings, but also improved the biological activity of the dressings. Sharaf et al. [151], prepared nanofiber dressings with different concentrations of bioactive glasses (BGs) based on CA. It was shown that nanofiber membranes loaded with BGs have high efficiency and rapid healing ability for diabetic wounds. Liu et al. [113], used zein (Zein) as a plasticizer to reduce the hydrogen bonding between CA and solvent. The results showed that the blended CA/Zein has excellent spinnability and the fiber membrane has favorable water absorption and swelling properties, which laid the foundation for controlled drug release. Furthermore, high concentrations of sesamol foster the expression of growth factors and induce the production of keratinocytes. Tan et al. [143], prepared pramipexole-containing CA fiber scaffolds and showed that the experimental group containing 3% pramipexole had higher collagen deposition and inflammation mitigation than the other experimental groups.

Cam et al. [116], reported an “all-natural” medicated wound dressing that loaded glibenclamide (GB) and metformin (Met) in gelatin and bacterial cellulose (BC) for electrospinning by portable electrospinning device, as shown in Figure 6A. Met is a member of the biguanide family and is a first-line hypoglycemic agent in the clinical therapy of patients with type 2 diabetes. The topical application of Met for diabetic wound therapy has been reported as effective [152]. GB is a sulfonylurea hypoglycemic medication that reduces blood-glucose levels by increasing insulin release from pancreatic cells and blocking hepatic glycogenolysis and glycogen xenobiotic effects [153]. Animal experiments and histological analysis demonstrated that the drug-loaded group had faster healing properties compared to the control group. At the end of the experiment, the greatest decrease in tumor necrosis factor (TNF-α) levels was observed in the gelatin/BC/GB treatment group, which was conducive to the transition of the wound from the inflammatory stage to the proliferative stage, thus effectively promoting wound healing [116]. The nanofiber membranes prepared in this research have high bioavailability and low side effects, and thus have important potential for application in diabetic wound healing.

### 4.2. Synthetic Polymer Electrospinning Fiber for Diabetic Wounds

Synthetic polymers are a rich variety of polymeric compounds that are synthesized artificially. These polymers have numerous advantages over natural polymers. They are widely used in the manufacture of electrospinning nanofiber membranes because they can be custom processed to obtain specific structures and properties, including essential mechanical capabilities, thermal stability properties, and outstanding spinnability [107,154].

Polycaprolactone (PCL) is a biodegradable polymer with a semi-crystalline structure, which has the advantages of non-immunogenicity, high mechanical strength, excellent spinnability, and good blending compatibility. It is extensively applied in tissue regeneration and wound healing applications because it facilitates faster wound healing and reduces the inflammatory response [155,156]. Therefore, PCL is an ideal material for the treatment of diabetic chronic wounds [157].

Polyvinyl alcohol (PVA) is a polymer obtained by the hydrolysis or alcoholysis of vinyl acetate, with high hydrophilicity, non-toxicity, and excellent processing properties. In diabetic wound healing, chronic wound healing was significantly promoted by the addition of NO to the PVA dressing. This is because the quality of granulation tissue was significantly improved, and wound strength increased through the release of NO. PVA can also improve water absorption permeability and the water-vapor transmission rate by cross-linking [29]. Therefore, PVA bioactive dressing shows a promising application in the therapy of diabetic chronic wounds.

Synthetic polymer electrospinning fibers have a distinctive structure, excellent mechanical properties, high thermal stability and favorable chemical stability, and can therefore be applied directly to the therapy of diabetic wounds. Mabrouk et al. [127], utilized electrospinning to create hierarchically structured fibrous-film trauma dressings, which consisted of polyacrylic acid (PAA) as a skin contact layer, polyvinylpyrrolidone (PVP) as an intermediate layer, and polycaprolactone (PCL) as the outermost layer, with the PVP layer loaded with the antibiotic ciprofloxacin (CIP). SEM results confirmed the fiber and membrane properties of layers with a nano to micron size range, and the mechanical properties of hierarchically structured fiber membrane creation exhibited a tensile strength of 12.8 ± 0.5 Mpa. Lee et al. [129], wrapped insulin in a core layer for diabetic wound therapy by coaxial electrospinning, and the fibrous membrane with core-shell structure had appropriate mechanical properties, and insulin was released in a controlled manner for 21 days due to the protective effect of the shell layer of PLGA. The addition of insulin decreases the level of type I collagen, increases the expression of growth factors and promotes vascular regeneration. Sun et al. [158], prepared a bionic scaffold by coaxial electrospinning to simulate the order of appearance of two collagens in wound healing. The bionic scaffold modulates the secretion of immune factors that trigger the conversion of pro-inflammatory macrophages (M1) to healing-promoting macrophages (M2). Antoine et al. [130], prepared zwitterionic electrospun nanofiber membranes of polyvinylidene fluoride (PVDF) and polystyrene and poly(4-vinyl pyridine) copolymer (zP(S-r-4VP)). Zwitterions facilitate the formation of a fibrous-membrane hydration layer and improve the high level of exudation in diabetic wounds, providing a rational healing environment for type 2 diabetic wounds.

### 4.3. Natural/Synthetic Polymer Hybrid Electrospinning Fiber for Diabetic Wounds

Although natural polymers have excellent biocompatibility, they exhibit poor mechanical properties in the body fluid environment, limiting their application to a certain extent. The blending of natural and synthetic polymers is regarded as a frontier strategy to overcome the limitations of natural polymers. The preparation of natural/synthetic polymer nanofibers by this method not only ensures the biological activity of natural polymers, but also improves the poor mechanical properties of natural polymers, which greatly expands the application of natural polymers [107,154].

Agarwal et al. [117], fabricated curcumin-loaded nanofibrous membranes with hydrophobic polycaprolactone (PCL) and hydrophilic polyvinyl alcohol (PVA) blended with filamentous protein (SF). The mechanical tensile strength of the curcumin composite fiber membrane was found to be 12.41–16.80 MP, which satisfied the criteria for an ideal trauma dressing. Liu et al. [159], prepared MgO/PCL/gelatin electrospun fibrous membranes. The results showed that the addition of MgO increased the pro-angiogenic properties and anti-degradation properties of the nanofiber membranes.

Davani et al. [94], Prepared dual drug-loaded nanofiber membranes with polyethylene oxide (PEO), Chitosan (CS) and vancomycin as the shell layer and gelatin (Gel), polyvinylpyrrolidone (PVP) and imipenem/cilastatin as the core layer by coaxial electrospinning. The obtained results demonstrate that the fiber membrane has a uniform and smooth core-shell structure, is non-cytotoxic and has a proper mechanical strength. Gabapentin (GBP) is an antiepileptic drug that effectively reduces neuropathic pain with minimal side effects [160]. Mostofizadeh et al. [95], employed coaxial electrospinning to prepare gelatin/PLGA-doped gabapentin and ciprofloxacin hydrochloride core-shell fiber. GBP in the shell is released particularly rapidly in the early stages of drug release, which facilitates the mitigation of acute neuropathic pain in diabetic wounds. Zhang et al. [161], developed a coaxial aqueous membrane that mimics the extracellular matrix. The study showed that this aqueous membrane could maintain a moist wound healing environment and regulate the expression of inflammatory cytokines. It could accelerate effective healing of diabetic wounds.

Derakhshan et al. [128], fabricated a multilayer electrospun fiber trauma dressing using PCL/collagen as the material, as shown in Figure 7D. The collagen layer containing herbal extracts not only facilitates angiogenesis, but also provides the necessary substrate for cell adhesion and proliferation. L929 fibroblasts displayed appropriate cell viability at varying drug loading doses in the fibrous membrane, as indicated by cell studies and MTT experiments. Zhang et al. [123], fabricated Janus nanofiber composite dressings loaded with curcumin by an electrospinning technique using quaternized chitosan (QCS) and polyvinyl alcohol (PVA) as substrates. Janus nanofiber composite dressing is asymmetrical and enables the autonomous unidirectional transmission of exudate, which well prevents the reverse penetration of exudate and avoids soaking of the outer edge of the ulcerated wound. Yu et al. [126], prepared a hierarchical nanofiber dressing with a hydrophobic outer layer and a wettable inner layer. The hydrophobic outer layer has antibacterial adhesion and reduces the risk of wound infection. The hydrophilic inner layer promotes vascular regeneration and collagen deposition to accelerate wound healing. It provides a new pathway for diabetic wound healing.

### 4.4. Electrospinning Fiber Multi-Drug Combination for Diabetic Wounds

At present, diabetic patients are clinically treated with drugs, however, one drug may not achieve the desired requirements, and the use of multiple drug combinations will often significantly improve the therapeutic effect. Multiple drug combinations include the combination of inorganic nanoparticles and natural drugs, growth factors or hypoglycemic agents and active drugs, etc.

#### 4.4.1. Inorganic Nanoparticle and Natural Drug Combination Promote Diabetic Wound Healing

Inorganic nanoparticles are widely applied in wound dressing applications due to their reduced side effects and the fact they do not easily cause microbial resistance [99,162]. Zinc oxide (ZnO) is widely applied in the medical field because of its excellent antibacterial activity, low price and non-toxicity [163,164]. Zinc’s biological effects on human and animal creatures have been studied extensively, and it has been shown that, in addition to its antibacterial and anti-inflammatory properties, zinc also functions as a trace element in the body and aids in the control of protein factor coding. Examples include transcription, extracellular matrix reconstruction, and regulation involved in the wound healing phase [165,166]. Thus, its capacity to facilitate vascular regeneration and re-epithelialization and modulate the expression levels of inflammatory factors makes Zn valuable in the therapy of diabetic wounds. Khan et al. [120], dual loaded both ZnONPs and oregano essential oil (OEO)-active substances, expecting to facilitate diabetic wound healing by increasing the anti-inflammatory effect, as shown in Figure 8A. Antibacterial experiments showed that fibrous membranes loaded with both active substances had higher antibacterial activity compared to those with a single-drug loading, illustrating that the combination of two antibacterial drugs with synergistic effects can be another option to avoid the toxic effects of high levels of antibacterial drugs and to reduce the development of bacterial resistance. Histological analysis and in vitro healing experiments also showed that the dual drug combination of fibrous membranes had more collagen deposition, neo-vascular regeneration, and faster healing cycles compared to the other groups.

Hussein et al. [137], fabricated fibrous dressings for diabetic wound treatment by a dual-jet electrospinning technique, using poorly degradable polyurethane (PU) and degradable polyvinyl alcohol (PVA) and gelatin (Gel) as substrates loaded with cinnamon essential oil (CEO) and nCeO_2_-active substances, respectively, as shown in Figure 8F. The main component of the CEO extracted from the bark of the plant is cinnamaldehyde, which has a widespread antibacterial effect. By introducing it into the electrospinning process, its disadvantage of poor water solubility can be improved. The composite fiber membranes exhibited appreciable water absorption and swelling, biodegradability and excellent mechanical strength. The presence of CEO significantly improved the loading of nCeO_2_ with higher antibacterial properties and cell viability compared to the blank control and experimental group loaded with nCeO_2_ alone.

Wang et al. [138], successfully synthesized a metal–organic framework eutectic with curcumin and zinc ions by the eutectic melting method and investigated the synergistic effects of hierarchical nanofiber-loaded eutectic metal–organic frameworks in the treatment of diabetic mouse wounds, as shown in Figure 8G. The results showed that a dual controlled-release system of curcumin and zinc ions can reduce the level of reactive oxygen species in the inflammatory phase of diabetic wounds and promote epithelial and vascular regeneration. This accelerated the transformation of the wound surface to the proliferative phase.

#### 4.4.2. Growth Factor-Loaded Electrospun Wound Dressing Promotes Diabetic Wound Healing

Platelet-derived growth factor (PDGF), angiogenic growth factors (bFGF), vascular endothelial growth factor (VEGF), transforming growth factor-β (TGF-β), and connective tissue growth factor (CTGF) together aid wound healing by encouraging neovascularization, epithelial tissue regeneration, and connective tissue repair. Nevertheless, in diabetic wounds, the overexpression of inflammatory cytokines causes the degradation of a significant amount of growth factors and prevents the complete healing of diabetic wounds. Therefore, the necessity of introducing growth factors in the therapy of diabetic wounds has increased considerably [167]. In addition, growth factors suffer from short half-life and poor stability, while the large surface area and high porosity structure of electrospun fibers could serve as an effective delivery system for growth factors.

Dwivedi et al. [168], developed a dual-loaded gentamicin sulfate (GS) and recombinant human epidermal growth factor (rhEGF) porous nanofiber dressing. The dressing showed high similarity to the extracellular matrix and had a porosity of up to 76%. The addition of rhEGF significantly facilitates the adhesion and proliferation of cells and scaffolds for the purpose of stimulating tissue regeneration. In an in vitro test, the GS/rhEGF drug-loaded membrane exhibited superior inhibition of *S. aureus*, *P. aeruginosa* and *E. coli* than the single-drug-loaded therapy group, with no significant side effects. In another study, Xie et al. [169], prepared a bionic fiber dressing with vascular endothelial growth factor (VEGF) and platelet-derived growth factor nanoparticles (PDGF-BB). This accelerates wound healing through the phased release of VEGF and PDGF-BB to enhance cell proliferation and collagen deposition. Amritha et al. [124], prepared functionalized nanofiber dressings containing dual growth factors. Through the synergistic effect of bFGF and VEGF, diabetic wounds successfully transition to the proliferative stage.

#### 4.4.3. Combination of Glucose-Lowering Drugs to Promote Diabetic Wound Healing

Glucose-lowering drugs lower blood sugar by restoring the insulin inhibition of adenylyl cyclase, inhibiting the intestinal absorption of glucose, reducing hepatic glucose output, improving insulin sensitivity, increasing glucose uptake and utilization by peripheral tissues, muscle and fat, and promoting anaerobic glycolysis. Recent clinical and animal research has confirmed that combination therapy with glucose-lowering drugs can effectively modify insulin resistance, oxidative stress and inflammatory response for the effective treatment of diabetes and its complications [170,171,172]. Therefore, the loading of hypoglycemic agents into fiber dressings for diabetic trauma therapy by electrospinning technology is of great significance.

Cam et al. [122], combined the oral hypoglycemic agents pioglitazone (PHR), metformin (MET) and glibenclamide by loading them in Gel/CS/PCL and PVP/PCL composite fibrous materials for the therapy of type 1 diabetic rat traumas in combination. As shown in Figure 9A, two types of fiber scaffolds were prepared by electrostatic spinning and pressurized gyration, respectively. Figure 9B,C shows the experimental results for diabetic rats with different control groups of wound healing appearance and wound area, respectively. All drug-loaded fibrous membrane wounds healed better compared to the unloaded fiber control group. Figure 9D shows the expression levels of inflammatory factors during wound healing in each experimental group. The combination-loaded PM3S group significantly down-regulated the expression of TNF-α, IL-6 and IL-1β pro-inflammatory factors and up-regulated the expression of IL-10 anti-inflammatory factors compared with other experimental groups. This facilitated the transition of diabetic wounds into the proliferative phase, thus effectively facilitating the effective healing of diabetic chronic wounds.

## 5. Conclusions and Future Prospects

Chronic diabetes wounds include various issues, including a protracted inflammatory cycle, angiogenesis difficulties, and bacterial infection. Not only does it bring great suffering to patients, but it also leads to a significant burden on the fragile healthcare systems of developing countries. Despite all of the efforts taken to manage it, the intended therapeutic outcomes have not been reached. Therefore, effective trauma management through an effective drug-delivery system is urgently needed. Nanofiber membranes prepared by electrospinning exhibit large specific surface area, high porosity, easy size control, easy surface functionalization and high similarity to the extracellular matrix. Therefore, they are considered advanced wound dressings. They show remarkable potential for encapsulation and the delivery of active substances that promote wound healing.

The development of diabetic chronic wound dressings faces a series of challenges, such as dressings used in clinical practice that cannot actively absorb and remove wound exudate, fail to retain wetness, easily adhere to the wound surfaces, induce an immune response, and have no active facilitation in the wound healing process. The preparation of nanofibrous membranes loaded with bioactive components by electrospinning can overcome these drawbacks and stimulate cell adhesion and migration, regulating the inflammatory process. They also facilitate the transformation of pro-inflammatory M1 phagocytes to anti-inflammatory M2 phagocytes, which provides a prerequisite for the healing of chronic diabetic wounds.

With the development of electrospinning technology, the process is also gradually developing from traditional single-fluid electrospinning to coaxial electrospinning, side-by-side electrospinning, and multifluid electrospinning, etc. However, it is necessary to investigate the process parameters in depth to develop a reproducible, stable, and safe nanofiber preparation method to ensure the mass production of diabetic wound dressings for nanofibers. In addition, it is difficult to achieve comprehensive treatment of diabetic chronic wounds by applying a single drug, therefore, the development of multi-drug combination therapy and complex structured nanofibers with multi-phase controlled-release of a single drug provides a promising strategy for chronic diabetic wound treatment.

## Figures and Tables

**Figure 1 biomolecules-12-01727-f001:**
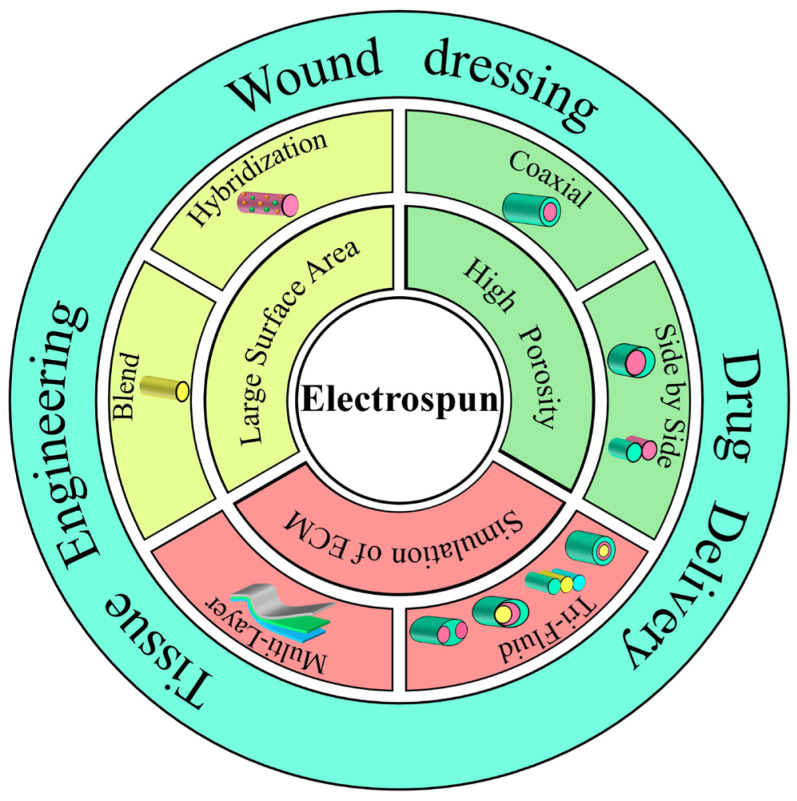
Characteristics, common structures of electrospun nanofibers and applications in the medical field.

**Figure 2 biomolecules-12-01727-f002:**
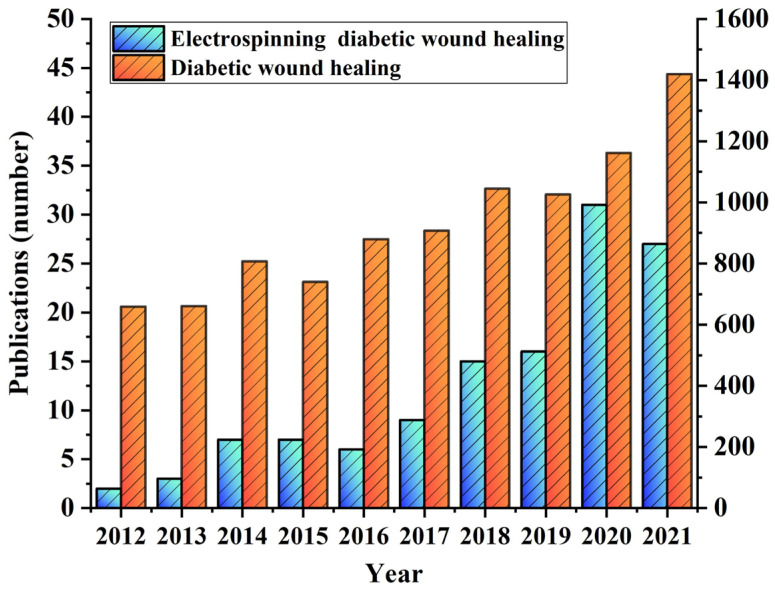
A number of articles published in the field of “Electrospinning diabetic wound healing” and “Diabetic wound healing” on the “Web of Science”, respectively.

**Figure 3 biomolecules-12-01727-f003:**
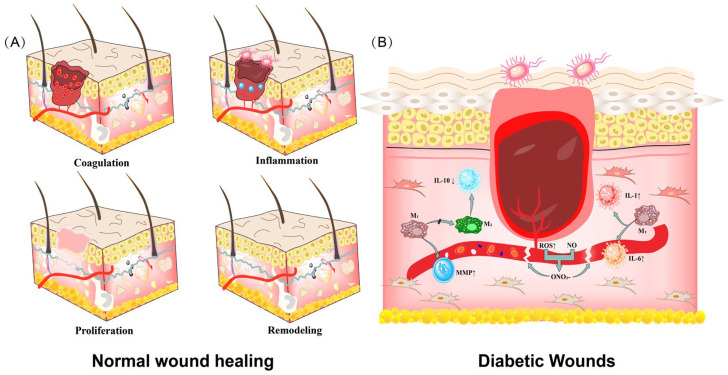
(**A**) Diagram of the normal wound healing process. (**B**) Diabetic wounds with overexpression of inflammatory factors. ROS reacts with NO to create peroxynitrite (ONO_2_^−^) in a high-glucose environment, impairing angiogenesis.

**Figure 4 biomolecules-12-01727-f004:**
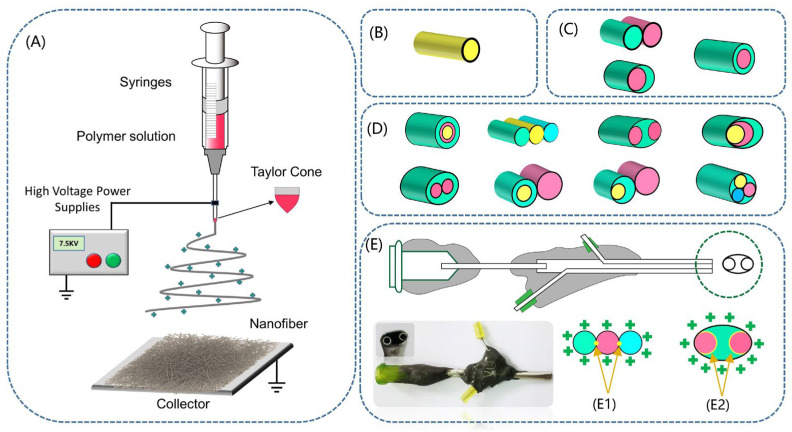
(**A**) Schematic diagram of electrospinning equipment. (**B**) Single-fluid electrospun technology. (**C**) Double-fluid side-by-side and coaxial electrospun. (**D**) Tri-fluid coaxial and tri-layer side-by-side and special structures electrospun technology. (**E**) Tri-layer side-by-side electrospinning needle schematic and realistic pictures.

**Figure 5 biomolecules-12-01727-f005:**
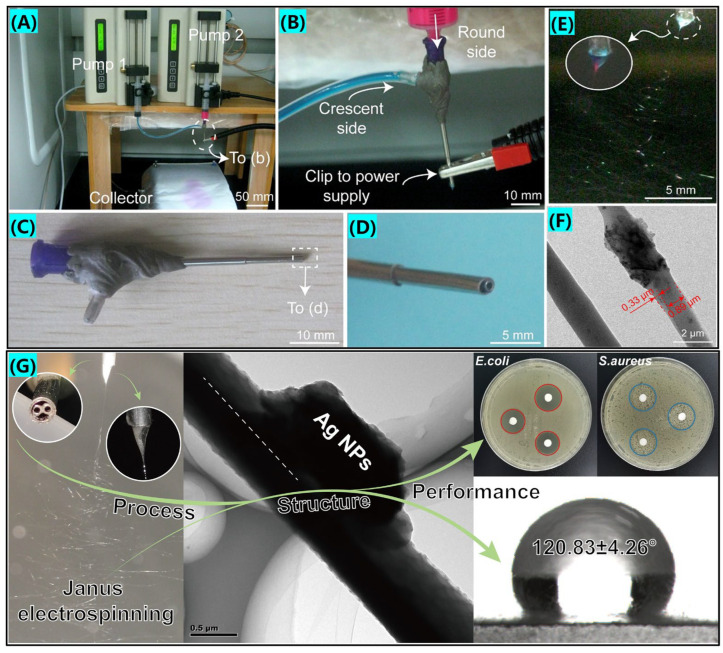
(**A**) Domestic electrospinning device, approved citation from ref. [100]. Copyright 2022, Elsevier. (**B**) Connection of the spinneret with working fluids and power supply, approved citation from ref. [100]. Copyright 2022, Elsevier. (**C**) Modified side-by-side electrospinning needle schematic realistic pictures, approved citation from ref. [100]. Copyright 2022, Elsevier. (**D**) Enlarged image of the top of the spinneret, approved citation from ref. [100]. Copyright 2022, Elsevier. (**E**) Modified side-by-side spinning for creating the Janus nanofiber, approved citation from ref. [100]. Copyright 2022, Elsevier. (**F**) TEM image of the Janus nanofiber, approved citation from ref. [100]. Copyright 2022, Elsevier. (**G**) Synergistic antibacterial study of organic-inorganic hybridized conventional Janus nanofibers.

**Figure 6 biomolecules-12-01727-f006:**
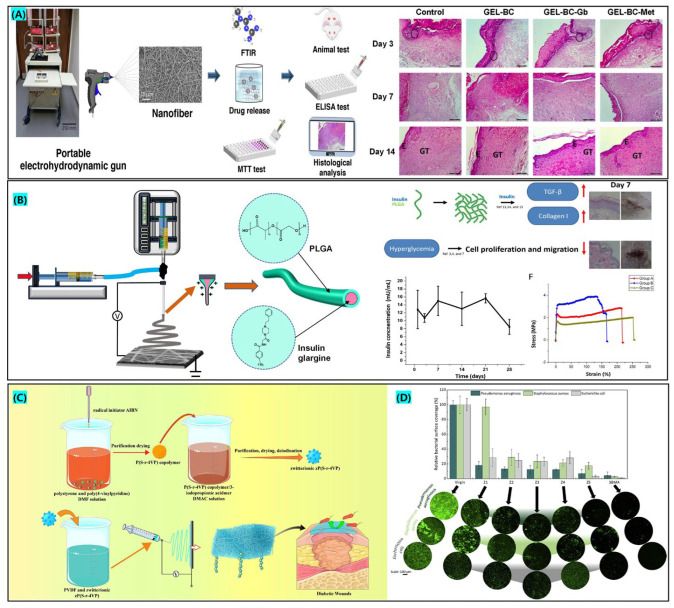
(**A**). Preparation of “all-natural” diabetic wound dressing by portable electrospinning, approved citation from ref. [116]. Copyright 2020, Elsevier. (**B**). Preparation of insulin-loaded core-shell nanofibers for diabetic wound therapy, approved citation from ref. [129]. Copyright 2020, Elsevier. (**C**). Synthesis of zwitterionic and preparation of nanofiber membranes through electrospinning. (**D**) Antibacterial properties of varying concentrations of zwitterionic nanofiber membranes, approved citation from ref. [130]. Copyright 2020, Elsevier.

**Figure 7 biomolecules-12-01727-f007:**
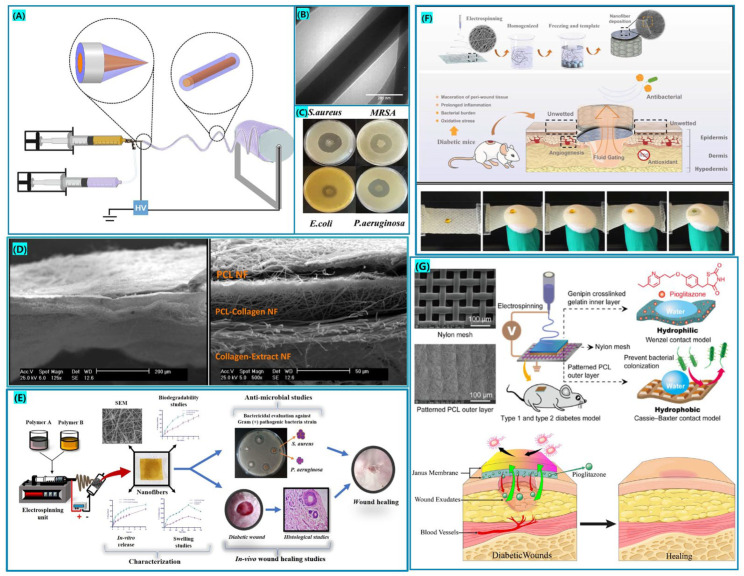
(**A**) Diagram of the fabrication of dual drug-carrying nanofiber membrane, approved citation from ref. [94]. Copyright 2021, Elsevier. (**B**) TEM image of the Core-shell nanofiber, approved citation from ref. [94]. Copyright 2021, Elsevier. (**C**) Antibacterial circle test of drug-laden core-shell fiber membrane against different bacteria, approved citation from ref. [94]. Copyright 2021, Elsevier. (**D**) Multilayer structured electrospun dressing SEM micrograph approved citation from ref. [128]. Copyright 2022, Springer Nature. (**E**) Polyvinyl alcohol (PVA)-sodium alginate (SA)-silk fibroin (SF)-cumene glycosides-loaded multifunctional nanofiber dressing. (**F**) Janus nanofiber dressing made from quaternary chitosan (QCS)/polyvinyl alcohol (PVA)-polycaprolactone (PCL)/loaded curcumin, approved citation from ref. [133]. Copyright 2022, Elsevier. (**G**) PCL/Gel-Pioglitazone nanofibrous membrane preparation for diabetic wound healing, approved citation from ref. [126]. Copyright 2020, American Chemical Society.

**Figure 8 biomolecules-12-01727-f008:**
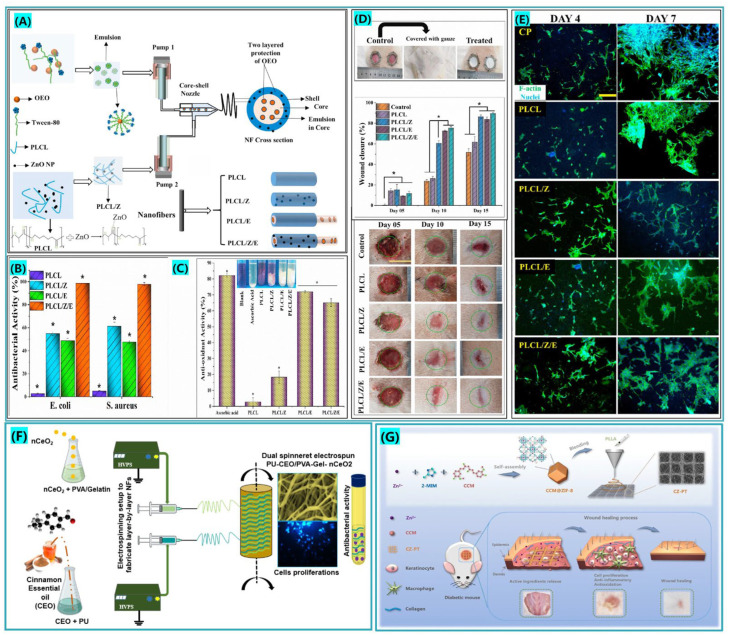
(**A**) Core-shell nanofibers containing ZnONPs nanoparticles and oil emulsions are created, approved citation from ref. [120]. Copyright 2021, Elsevier. (**B**) The antibacterial activity of each set of fiber membranes was tested, approved citation from ref. [120]. Copyright 2021, Elsevier. (**C**) Study of antioxidant activity of fibrous membranes, approved citation from ref. [120]. Copyright 2021, Elsevier. (**D**) Various nanofiber membranes are used to cure diabetic wounds. Statistical analysis was performed by one-way ANOVA, approved citation from ref. [120]. Copyright 2021, Elsevier. (**E**) F-actin/DAPI staining was used to examine the morphology of adherent cells, approved citation from ref. [120]. Copyright 2021, Elsevier. (**F**) Schematic diagram of double spinneret electrospinning preparation of nanofibers loaded with cinnamon essential oil and nCeO_2_. (**G**) Eutectic metal-organic backbone (MOF) composite nanofiber scaffold containing curcumin and zinc ions, approved citation from ref. [138]. Copyright 2020, Elsevier. * *p* < 0.05.

**Figure 9 biomolecules-12-01727-f009:**
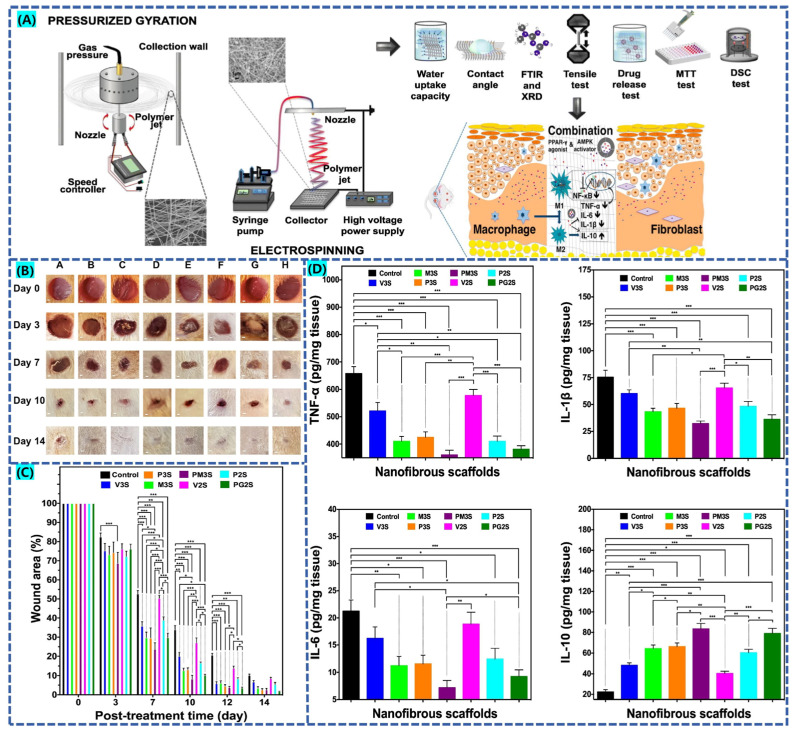
(**A**) Fabrication of drug-loaded nanofiber membranes by different methods, approved citation from ref. [122]. Copyright 2021, Elsevier. (**B**) Statistics on the appearance of wound healing in diabetic rats. (**C**) Results of wound closure experiments in different control groups, approved citation from ref. [122]. Copyright 2021, Elsevier. (**D**) The inflammatory factor levels in the control group and each drug-loaded experimental group. The data is presented as mean ± standard error of the mean. * *p* < 0.05, ** *p* < 0.01, *** *p* < 0.001, approved citation from ref. [122]. Copyright 2021, Elsevier.

**Table 1 biomolecules-12-01727-t001:** Summary of applications on the application of conventional polymeric materials generated by electrospinning to improve diabetic wound healing.

Polymer	Additional Polymer	Solvent	Active Ingredients	Technique	Highlights	Ref.
CA	Gel	HFIP	Berberine	Blend	Antimicrobial research on wounds has shown that they effectively suppress the microbial growth.	[112]
	Zein	Acetic acid, water	Sesamol	Blend	Synergistic effect of sesamol and nanofibrous membranes.	[113]
	PEO/SF	Acetone, Formic acid	MB, CIP	Blend	ROS generation during light-emitting diode irradiation, achieved combined PDT and antibiotic therapy.	[114]
Gel	PVA	DMSO	Ceph	Blend	Stronger antibacterial activity and thermal stability, drug loaded fibrous membranes have a dual purpose.	[115]
	PLGA	DMF Methanol TCM, DIW	GBP, CipHCl	Coaxial	Core-shell fibers for dual delivery of analgesic and antibiotic agents for the therapy of diabetic neuropathic ulcers.	[95]
	BC	Acetic acid, DMF, DIW	MET, GB	Blend	An “all-natural” drug-eluting wound dressing. Gel-BC-GB significantly reduces the expression of TNF-α.	[116]
SF	PCL/PVA	DCM, Formic acid	Curcumin	Blend	Synthetic and natural polymer blends and active ingredients are encapsulated for electrospinning.	[117]
	/	/	Hydroxyapatite, Curcumin	Blend	The mechanical characteristics and porosity of the fiber membrane improved when the CUR concentration was raised.	[118]
HA	PLGA	HFIP, DIW	EGCG	Coaxial	The synergistic impact of HA and EGCG increased wound epithelialization and ECM rearrangement considerably.	[119]
	PLCL	HFIP	ZnO, OEO	Coaxial	Controlled release of the volatile chemical OEO at the core of the fiber in the form of an oil-in-water emulsion.	[120]
	PLA	DCM	Valsartan	Blend	Composite fibers have a dynamic effect and promote re-epithelialization.	[121]
CS	Gel/PCL/PVP	Acetic acid DIW, TCM Methanol	MET, PHR, GB	Three-layered	Multi-drug combination therapy accelerates diabetic wound healing in type 1 diabetic rats.	[122]
	PVA/PCL	Acetic acid, DCM, DMF	Curcumin	Janus	Enables autonomous, rapid and unidirectional transfer of exudate, effectively preventing reverse penetration of liquids.	[123]
	Collagen/PLGA	HFIP, TFA	VEGF, bFGF	Blend	The collagen gene expression in composite growth factor fiber dressing is greater, and the dressing heals faster.	[124]
	PVA	Methanol, Acetic acid, DIW	ZnO	Blend	The composite nanofiber membrane has a 90% closure rate for diabetic wounds.	[125]
PCL	Gel	HFIP	Pioglitazone	Blend	The scaffolds increased epidermal regeneration, angiogenesis, collagen deposition, and inflammatory responses in vivo. Synergistic improvement of diabetic wound healing efficiency	[126]
	PAA/PVP	Ethanol, DCM	CIP	Three-layered	Excellent mechanical properties.	[127]
	Collagen	Acetic acid, Methanol, TFM	Melilotus	Three-layered	Each layer of the structure has unique properties that synergistically promote wound healing	[128]
PLGA	/	HFIP	Insulin	Coaxial	The topical application of insulin reduces the amount of type I collagen in vitro and increases the amount of TGF-β in vivo, promoting the healing of diabetic wounds.	[129]
PVDF	ZP (S-r-4VP)	Acetone, DMF, DMAC	/	Blend	The amphoteric copolymer membrane possesses high hemocompatibility, non-adhesive characteristics, and resistance to contamination by plasma proteins.	[130]
PVA	PCL	Ultrapure water DMF, DCM	SDF1	Blend	SDF1-loaded fiber membranes displayed a high level of cellular activity and enhanced cell proliferation considerably.	[131]
	/	Acetic acid	EPP	Blend	Outstanding water absorption performance, inflammatory response suppression, and wound healing time reduction.	[132]
	SA/SF	Formic acid, Calcium chloride DIW	Asiaticoside	Blend	Low toxicity and considerable cell migration, restoring the skin’s natural capacity to recover.	[133]
PLA	PVA	DIW DMF, DCM	CTGF	Coaxial	A core-shell membrane containing CTGF promotes cell proliferation, migration, and angiogenesis, allowing for faster wound healing.	[134]
	PVA	DMF, DCM DIW	nCeO_2_	Three-layered	The hierarchical structure may be used as a medication carrier system that also promotes cell migration and proliferation.	[135]
	CS	DMF, DIW	Cod liver oil	Blend	Provides the permeability and oxygen essential for tissue healing.	[136]
PU	PVA/Gel	THF, DMF	nCeO_2_, CEO	Blend	CEO promotes antibacterial effectiveness and cell viability by increasing nCeO_2_ loading.	[137]
PLLA	/	Acetone DMF, DCM	Curcumin, Zn^+2^	Blend	During the wound healing phase, the multilayered nanofiber scaffold releases curcumin and zinc ions as needed.	[138]

Note: (MB) methylene blue, (CIP) ciprofloxacin, (ROS) reactive oxygen species, (PDT) Photodynamic therapy, (DIW) distilled water, (GBP) gabapentin, (CipHCl) ciprofloxacin hydrochloride, (Met) metformin, (GB) glibenclamide, (TNF-α) Traumatic tumor necrosis factor α, (DCM) dichloromethane, (PAA) polyacrylic acid, (Ceph) cephalosporin antibiotic, (HFIP) 1,1,1,3,3,3-hexafluoroisopropanol,(EGCG) epigallocatechin-3-O-gallate, (PLCL) Poly (L-lactide-co-caprolactone), (OEO) Oregano essential oil, Trifluoroacetic acid (TFA), Vascular endothelial growth factor (VEGF), Basic multicellular growth factor (bFGF), Dimethyl sulfoxide (DMSO), (PRP) Platelet Rich Plasma Gel, (PHR) Pioglitazone, transforming growth factor-β (TGF-β), Stromal cell-derived factor 1 (SDF1), Enteromorpha polysaccharide (EPP), Connective tissue growth factor (CTGF), ZP (S-r-4VP) -polystyrene and poly (4-vinylpyridine) copolymer.

## Data Availability

Not applicable.

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
