# Peer review of "Advances in the Preparation of Nanofiber Dressings by Electrospinning for Promoting Diabetic Wound Healing"

_biomolecules, 2022, doi:10.3390/biom12121727_

Round 1

Reviewer 1 Report

In this review, authors summarized recent advances of nanofiber fabricated by electro-spinning and their applications in promoting diabetic wound healing. The review is well organized with detailed discussions. I recommend its publishing on Biomolecules. Please see some comments below:

1. Figure 1 is not easy to understand.

2. It seems the introduction of background, like section 2 and section 3, takes too much space. In reviewer's opinion, section 4 should be highlighted with more discussions.

3. Section 5 Conclusion and Future Prospects should be expanded.

Author Response

Dear Reviewer,

Thank you very much for giving us the precious opportunity to revise the manuscript and your kind guidance for submitting it again! We are very grateful to your pertinent comments and suggestions. Accordingly, we carefully revised our manuscripts and provided our responses to each comment. In the revision, all revisions are highlighted in revision mode. Thank you again!

We revised the comments of reviewers, and the results are as follows. (Words in black are comments & in blue are responses).

1. Figure 1 is not easy to understand.

Response: Many thanks for your comments, Figure 1 mainly depicts the advantages of electrospinning to prepare nanofibers. Commonly used structures are Single spinning, hybrid, coaxial, side-by-side, three-fluid and multilayer structures. and applications in the biomedical field. We made changes below Figure 1.

2. It seems the introduction of background, like section 2 and section 3, takes too much space. In reviewer's opinion, section 4 should be highlighted with more discussions.

Response: Many thanks for your kind words and valuable suggestions! We have cut and added unnecessary parts of the article and tried to make the article more concise.

3. Section 5 Conclusion and Future Prospects should be expanded.

Response: Yes, many thanks! We have expanded the conclusions and future outlook sections to read as follows:

“However, it is necessary to investigate the process parameters in depth to develop a reproducible, stable, and safe nanofiber preparation method to ensure the mass pro-duction of diabetic wound dressings for nanofibers. In addition, it is difficult to achieve comprehensive treatment of diabetic chronic wounds by applying a single drug, therefore, the development of multi-drug combination therapy and complex structured nanofibers with multi-phase controlled release of single drug provides a promising strategy for chronic diabetic wound treatment.”

Thank you again for your valuable time and effort spent on our manuscript. With your precious helps, we believe that the quality of our manuscript has been improved. We sincerely hope that our manuscript is now suitable for publication in Biomolecules

Reviewer 2 Report

The article is detailed but there are some points to be clarified.

Introduction

The introduction on diabetes in general is too long. Please focus  on chronic diabetic ulcers.

Please specify the concept of tissue repair and regeneration

Neuropathy alteration arteropathy glycemia callositis in the pathogenesis of wounds of the diabetic foot

The gold standard therapies  for diabetic foot ulcer are:debridement , glycemic control,  dressings based on the characteristics of the wound and periwound skin. pressure reduction and use of orthotics.

Emphasize the importance of adequate cleansing, debridement and maintenance debridement. (please add a paragraph).

Basic dressings

Local treatment depends on the etiology of the ulcer and the characteristics of the wound bed and periwound skin.

Why did you only consider traumatic wounds if the topic is chronic diabetic wounds?

The dressings are chosen according to the principles of wound bed preparation and the acronym TIME.

The key concept is to use a moist environment to stimulate healing.

Please improve the dressing classification.

It is advisable to classify them in dressings that perform autolytic debridement (hydrogel hydrocolloids)

Enzymatic debridement (collagenase), antiseptic dressings, dressings that control exudate (hydro-fiber, alginates (which also have a haemostatic action) and polyurethane foams... and more

Antimicrobial dressings

Topical antibiotics are not commonly used to prevent infection because they cause bacterial resistance and contact dermatitis.

Better to use the active ingredient (PHBM) not the trade name (prontosan)

Please add all antimicrobid dressings.(for example iodine.)

There are also binding bacteria dressings that capture bacteria without releasing specific substances.

Author Response

Dear Reviewer,
Thank you very much for giving us the precious opportunity to revise the manuscript and your kind guidance for submitting it again! We are very grateful to your pertinent comments and suggestions. Accordingly, we carefully revised our manuscripts and provided our responses to each comment. In the revision, all revisions are highlighted in revision mode. Thank you again!
We revised the comments of reviewers, and the results are as follows. (Words in black are comments & in blue are responses).
1.The introduction on diabetes in general is too long. Please focus on chronic diabetic ulcers. Please specify the concept of tissue repair and regeneration Neuropathy alteration arteropathy glycemia callositis in the pathogenesis of wounds of the diabetic foot The gold standard therapies for diabetic foot ulcer are: debridement , glycemic control, dressings based on the characteristics of the wound and periwound skin. pressure reduction and use of orthotics. Emphasize the importance of adequate cleansing, debridement and maintenance debridement. (please add a paragraph).
Response: Yes, Thank you very much for your suggestion. We have removed the diabetes-related presentation within the article and focused on the direction of diabetic ulcers. The additions are in lines 45-64, 135-151 of the article. Concepts on tissue repair and proliferation have been added to lines 35-38 of the article.
2.Basic dressings
Local treatment depends on the etiology of the ulcer and the characteristics of the wound bed and periwound skin. Why did you only consider traumatic wounds if the topic is chronic diabetic wounds? The dressings are chosen according to the principles of wound bed preparation and the acronym TIME. The key concept is to use a moist environment to stimulate healing. Please improve the dressing classification. It is advisable to classify them in dressings that perform autolytic debridement (hydrogel hydrocolloids) Enzymatic debridement (collagenase), antiseptic dressings, dressings that control exudate (hydro-fiber, alginates (which also have a haemostatic action) and
polyurethane foams... and more
Response: Yes, thank you very much for the heads up! Sorry for that our original writing did lead to some ambiguity. This article considers the study of chronic ulcer wounds, and we have corrected the term traumatic wounds in the article. We then reclassified the base dressings, the main contents of which are as follows:
“(1) Autolytic debridement dressing: Autolytic debridement is a painless and highly selective primitive reaction of the body to eliminate necrotic material, often requiring the application of new dressings to create a moist environment to encourage necrotic tissue autolysis. The accompanying hydrogels and hydrocolloid dressings are widely used in clinical practice. Hydrogel is a wet wound dressing with good moisturizing ability, but it is easy to leak when using amorphous hydrogel for debridement, which affects the debridement effect. Hydrocolloids are wound dressings consisting of a cross-linked polymer matrix and an adhesive. The ability to provide a moderate cushioning effect makes it suitable for joint wounds, superficial ulcers and shallow leg ulcers. However, they are also prone to symptoms such as allergic skin reactions or adhesions, and the opaque nature of the dressing limits routine wound examination.(2)Enzymatic debridement dressing: Enzymatic debridement is a method that uses chemical reagents to identify and break down inactivated tissue, usually in smaller wounds and wounds with selective microenvironments that do not require intervention. Enzymatic debridement dressing has the advantages of being cost effective and efficient[48,49]. (3)Antiseptic dressings: Some research suggests that debridement alone is not sufficient to accelerate chronic wound healing and recommends the use of antiseptics to complement the debridement process and control infection. Antiseptics are a class of drugs that can prevent the growth or damage of microorganisms in living tissues or on the surface of living tissues. Halogenated compounds, alcoholbased agents, and biguanides are often used as antiseptics primarily for chronic wound care because they effectively inhibit bacterial reproduction and promote resistance to the external environment. (4) Alginate dressing: Alginate dressing consists of alginate salts, which are highly absorbent and hemostatic and suitable for highly exuding wounds. It is
clinically indicated for deep pressure ulcers of the lower extremities, noma, and exudative ulcers of the lower extremities. However, on wounds with little to no exudate, they adhere to the surface of the healing wound, causing pain and damaging healthy tissue when removed. (5) Polyurethane foams dressings are hydrocolloid dressings with moisturizing, soft, non-toxic and good mechanical proper-ties, and can be used as wound dressings even after complete immersion. In addition, the porous structure inside the polyurethane foam dressing makes the interconnected pores have the advantages of good intracellular growth, strong water absorption capacity, and high water vapor transmission rate. This allows it to effectively absorb exudate and maintain a moist microenvironment on the wound, accelerating wound healing and preventing wound infestation.”
3.Antimicrobial dressings
Topical antibiotics are not commonly used to prevent infection because they cause bacterial resistance and contact dermatitis. Better to use the active ingredient (PHBM) not the trade name (prontosan) Please add all antimicrobid dressings.(for example iodine.) There are also binding bacteria dressings that capture bacteria without releasing specific substances.
Response: Yes, many thanks! The main active ingredient of prontosan gel is polyhexamethylenebiguanide (PHBM), which we have corrected. We are sorry that we did not notice that antibiotics can cause bacterial resistance and contact dermatitis. We have removed antibiotic dressings from the article and added Iodine-based dressings.
Thank you again for your valuable time and effort spent on our manuscript. With your precious helps, we believe that the quality of our manuscript has been improved. We sincerely hope that our manuscript is now suitable for publication in Biomolecules and look forward to hearing from you at your earliest convenience.
With our sincerest regards,
Ke Wang

Reviewer 3 Report

 The manuscript entitled “Functionalized Electrospun Blended PCL/Gelatin – Graphene Oxide Nanofibers: Physicochemical Properties and Antibacterial Activity" by Nazirah Hamdan contains several interesting findings, and it may ultimately be suitable for publication. A significant effort was made to prepare the article, which is quite interesting and unique. This article is impressive for the reviewer and audience of the nanotechnology community as well as material science. This article would show a significant impact on the tissue engineering, nanomedicine, and materials science community. There are some suggestions to improve the quality of the manuscript as it stands (detailed below) and these need to be addressed before it can be considered further. I thus recommend the paper be accepted after a minor revision.

The reviewer has the following comments  

1.     Recently, the following comprehensive review article was published based on nanofibers

https://doi.org/10.1021/acsbiomaterials.2c00786

The authors should justify why there is room for another review

2.     The introduction section is very brief and may be improved entirely so that the reader can identify the scientific problems solved by this research. Moreover, the information on biomaterials may be elaborated on in the introduction with recent references (preferably 2021-2022). There are several biocompatible and biodegradable drug delivery systems (Nanofibers, hydrogel nanoparticles, hydrogels, thin films, etc.)-based on biomaterials but why did the authors select only nanofibers? Moreover, the information on nanofibers may be explored in the introduction with recent references, Thus, the following article(s) may be quoted in the introduction and other relevant sections.

https://doi.org/10.1021/acsbiomaterials.2c00786

It would be more realistic to cover such kind of research work in the current manuscript. Which will enrich the quality of the current manuscript as well as the inquisitiveness of the readers.

Author Response

Dear Reviewer,
Thank you very much for giving us the precious opportunity to revise the manuscript and your kind guidance for submitting it again! We are very grateful to your pertinent comments and suggestions. Accordingly, we carefully revised our manuscripts and provided our responses to each comment. In the revision, all revisions are highlighted in revision mode. Thank you again!
We revised the comments of reviewers, and the results are as follows. (Words in black are comments & in blue are responses).
1. Recently, the following comprehensive review article was published based on nanofibershttps://doi.org/10.1021/acsbiomaterials.2c00786.The authors should justify why there is room for another review
Response: Many thanks for your kind words and valuable suggestions! In this study, we summarize in detail the research of electrospun nanofiber dressings in the field of diabetic wounds based on natural and synthetic polymers from the perspective of coblended nanofibers, hybrid nanofibers, core-shell electrospun nanofibers and multilayer nanofibers, respectively. The strategy of using multidrug combinations for diabetic wounds is also highlighted. We sincerely hope to get your approval.
2. The introduction section is very brief and may be improved entirely so that the reader can identify the scientific problems solved by this research. Moreover, the information on biomaterials may be elaborated on in the introduction with recent references (preferably 2021-2022). There are several biocompatible and biodegradable drug delivery systems (Nanofibers, hydrogel nanoparticles, hydrogels, thin films, etc.)-based on biomaterials but why did the authors select only nanofibers?
Response: Thank you very much for your suggestion. We have expanded the
introductory section to hopefully better help the reader to understand the scientific questions addressed by this research. The citation information on biomaterials has been updated to the most recent literature. Although drug delivery systems such as films, hydrogels, and ointments have been used in the management of diabetic chronic wounds, nanofibers have been shown to be more effective in promoting tissue formation, vascular regeneration, and faster wound healing. The nanofibers have a structure similar to the natural extracellular matrix, which facilitates the adhesion of cells. In addition, the porous structure of nanofibers can enhance drug loading and provide continuous drug delivery, and the higher specific surface area is beneficial for hemostasis.
3. Moreover, the information on nanofibers may be explored in the introduction with recent references, Thus, the following article(s) may be quoted in the introduction and other relevant sections.https://doi.org/10.1021/acsbiomaterials.2c00786. It would be more realistic to cover such kind of research work in the current manuscript. Which will enrich the quality of the current manuscript as well as the inquisitiveness of the readers.
Response: Thank you very much for recommending this article to us. We have learned a lot of effective information about nanofibers from this article and have citing it into our article. Thank you very much for your suggestions!
With our sincerest regards,
Ke Wang
